# The Temporal and Geographical Dynamics of Potato Virus Y Diversity in Russia

**DOI:** 10.3390/ijms241914833

**Published:** 2023-10-02

**Authors:** Viktoriya O. Samarskaya, Eugene V. Ryabov, Nikita Gryzunov, Nadezhda Spechenkova, Maria Kuznetsova, Irina Ilina, Tatiana Suprunova, Michael E. Taliansky, Peter A. Ivanov, Natalia O. Kalinina

**Affiliations:** 1Shemyakin-Ovchinnikov Institute of Bioorganic Chemistry of the Russian Academy of Sciences, 117997 Moscow, Russia; viktoriya.samarskaya2012@yandex.ru (V.O.S.); nikgr1@yandex.ru (N.G.); rysalka47@gmail.com (N.S.); mariasha.k.1012@gmail.com (M.K.); irinailina.bio@gmail.com (I.I.); michael.taliansky@mail.ru (M.E.T.); 2Department of Entomology, University of Maryland, College Park, MD 20742, USA; eugene.ryabov@gmail.com; 3Faculty of Bioengineering and Bioinformatics, Lomonosov Moscow State University, 119234 Moscow, Russia; 4Doka-Gene Technologies Ltd., 141880 Rogachevo, Russia; suprunova@gmail.com; 5Faculty of Biology, Lomonosov Moscow State University, 119234 Moscow, Russia; regaflight@gmail.com; 6Belozersky Institute of Physico-Chemical Biology, Lomonosov Moscow State University, 119234 Moscow, Russia

**Keywords:** plant RNA virus, Potyvirus, *Solanum tuberosum*, plant–virus interactions, virus evolution, phylogeography, RNA virus recombination, virus transmission

## Abstract

Potato virus Y, an important viral pathogen of potato, has several genetic variants and geographic distributions which could be affected by environmental factors, aphid vectors, and reservoir plants. PVY is transmitted to virus-free potato plants by aphids and passed on to the next vegetative generations through tubers, but the effects of tuber transmission in PVY is largely unknown. By using high-throughput sequencing, we investigated PVY populations transmitted to potato plants by aphids in different climate zones of Russia, namely the Moscow and Astrakhan regions. We analyzed sprouts from the tubers produced by field-infected plants to investigate the impact of tuber transmission on PVY genetics. We found a significantly higher diversity of PVY isolates in the Astrakhan region, where winters are shorter and milder and summers are warmer compared to the Moscow region. While five PVY types, NTNa, NTNb, N:O, N-Wi, and SYR-I, were present in both regions, SYRI-II, SYRI-III, and 261-4 were only found in the Astrakhan region. All these recombinants were composed of the genome sections derived from PVY types O and N, but no full-length sequences of such types were present. The composition of the PVY variants in the tuber sprouts was not always the same as in their parental plants, suggesting that tuber transmission impacts PVY genetics.

## 1. Introduction

Potato virus Y (PVY) is the most economically important viral pathogen of potato (*Solanum tuberosum* L.). The virus is also a serious threat for tobacco, pepper and, to a lesser extent, tomato [1]. PVY is a picorna-like virus, a type representative of the genus *Potyvirus* (family *Potyviridae*), which currently consists of more than 190 species [2]. The PVY genome is represented by single-stranded, positive-sense RNA with a length of approximately 9.7 kb containing a poly(A) tail at the 3′ terminus and a covalently linked VPg protein at the 5′ terminus and coding for a large polyprotein that is cleaved by virus-specific proteases into ten mature proteins [3]. Trans-frame protein P3N-PIPO is produced via transcriptional slippage using an additional short open reading frame [4]. RNA is packed into flexible helical particles, with a length of about 740 nm and a width of 11 nm [3]. The virus is transmitted horizontally by aphids in a nonpersistent manner and vertically between vegetative generations through potato tubers. Aphid transmission increases the efficiency of the spread of the virus, especially in warm climates [5]. Solanaceous weeds can serve as a natural reservoir for the virus [6].

To date, several main strains of PVY are known. First of all, O (ordinary), N (necrotic), and C (common) strains that differ in biological properties should be mentioned. Strains O and C cause necrotic spots on leaves in potato plants with resistance genes *Ny* and *Nc*, respectively. As a rule, infection with strain N leads either to the appearance of mild symptoms on potato leaves or their absence; at the same time, it causes systemic necrotic symptoms in tobacco [5]. Phylogenetic analysis of full-sized genomes revealed five phylogroups [7] and principally confirmed the classification based on the interaction with the host plants. Lineage N is mainly represented by South American isolates and has spread to the rest of the world relatively recently. Isolates belonging to lineage O are distributed predominantly outside of South America. Lineage C is a branch of O; isolates are often found in Solanaceous hosts other than potato [8]. Strain N is subdivided into NA-N and Eu-N genotypes and strain O includes the distinct subgroup O5 [3,9]. Recombinants between strains N and O formed phylogroups R1 and R2 [8]. In recent years, recombinant strains that are capable of overcoming host resistance have largely replaced conventional ones in Europe and the US [10]. Among them, PVY-NTN and PVY-N-WI are the most studied [3,11]. Recombinant infections may be mild or asymptomatic, but lead to potato tuber necrotic ringspot disease, PTNRD (PVY-NTN), or tuber cracking (PVY-N-WI^i^) [3,12]. PVY-N:O recombinant is characterized by a single recombinant junction (RJ), while PVY-N-WI is characterized by two and PVY-NTN by three (NTNa) or four (NTNb) RJs that are comparatively conserved in the genome [3]. In addition to nine “common” recombinants (N:O, N-Wi, NTNa, NTNb, NE11, E, SYR-I, -II, and -III), some “rare” and “unclassified” ones were discovered [13].

PVY infection is sensitive to environmental conditions [14]. For example, the *Ny* resistance gene from *S. sparsipilum* and *S. sucrense* as well as *Ny-1* from potato only operate effectively at cool temperatures (16–20 °C). Otherwise, the infection becomes systemic [15]. Presumably, this effect is related to the conformational loss of function of the corresponding proteins [16]. According to some data, the efficiency of RNA interference (RNAi) increases at higher temperatures [17]. On the contrary, HC-Pro, the PVY suppressor of RNAi, is downregulated under such conditions [18]. The main enzymes of the methionine cycle are also affected by the temperature in potato cultivar Chicago, thus leading to the outbreak of infection [19]. Heat shock proteins positively influence the reproduction of PVY and other potyviruses [20,21,22].

In this study, we used high-throughput sequencing to characterize the diversity of PVY transmitted by aphid vectors to virus-free potato plants in the Moscow and Astrakhan regions of Russia (located approximately 1300 km apart), which have different climates, ranges of aphid vectors, and potential reservoir plants. We then analyzed PVY populations in the potato sprouts from the field-grown tubers. Our study identified the major PVY variants circulating in two regions of Russia and provided a novel insight into the effect of tuber transmission on PVY genetics.

## 2. Results

### 2.1. Field Spread and Tuber Transmission of PVY

In this study, we sought (i) to comprehensively characterize the diversity of PVY variants circulating in two areas of Russia with distinct climate characteristics, the Astrakhan and Moscow regions, and (ii) to determine whether the selection of PVY variants occurs during tuber transmission of the virus. The experiments included planting commercial virus-free seed potatoes produced in an insect-proof greenhouse of the PVY-susceptible varieties La Strada and Indigo in the years 2021 and 2022 in open fields in the Astrakhan and Moscow regions, and growing plants without insecticide treatment, thereby allowing for the natural transmission of PVY by aphids. The leaf samples were collected from the field-grown plants 10 to 12 weeks after planting when the plants were at the tuber formation stage. We also collected leaf samples of newly sprouted plants, 5 weeks after planting. The tubers were collected from the experimental aphid-exposed field-grown potato plants 3–4 weeks after leaf sampling. To produce the leaf material of the next vegetative generation, apical buds were excised from the tubers and sprouted in an insect-proof greenhouse. The samples from the field-grown plants and from the tuber sprouts were analyzed with RNA-seq to characterize the PVY strain composition and loads of the virus (Figure 1).

We obtained 42 samples, each derived from 10 to 25 plants, which were then used to produce RNA-seq libraries containing 28 to 109 million paired-end reads. To estimate the levels of PVY, we aligned the NGS reads to a reference containing a representative set of full-length genomic sequences of all known types of PVY and determined the proportions of PVY reads in the NGS libraries (Figure 2). We found that all leaf samples from the plants collected 10 to 12 weeks after planting were infected with PVY (Figure 2, “Leaves”, Appendix A), presumably as a result of exposure to viruliferous aphids. There were 0.17% to 4.50% of PVY reads in the NGS libraries from the samples. The plants which were sampled at a young stage, 5 weeks after tuber planting, had no PVY reads (Figure 2, “Leaves young plants”, Appendix A), which further confirmed that the seed potatoes used in the experiments were virus-free and all PVY was acquired via field-grown potato plants as a result of aphid transmission. The virus infection was transmitted through tubers to the next vegetative generation. The levels of PVY varied considerably, with the proportion of PVY reads in the NGS libraries from the tuber sprouts ranging from 0.18% to 7.32% (Figure 2, “Tubers”, Appendix A).

Although both potato varieties used in the experiment were susceptible to PVY, it was reported that Indigo potatoes are more susceptible to PVY compared to La Strada. Indeed, we found that the number of PVY reads were 1.4–6.2-fold higher in the NGS libraries from the leaf material from fully grown Indigo plants compared to that from La Strada in both years, and this difference was statistically significant for the 2021 Moscow region samples (*p* = 0.0007, ANOVA). Similarly, the loads of PVY were higher in Indigo tuber sprouts compared to La Strada for both years and regions. In the year 2021, this difference was statistically significant for both the Moscow region (1.4-fold, *p* = 0.0349, ANOVA) and the Astrakhan region (1.8-fold, *p* = 0.0113, ANOVA). Since there were no biological repeats for the treatments in the year 2022, it was not possible to assess the statistical significance of the PVY load differences.

We observed a significant positive correlation between the loads of PVY in the leaves of the field-grown potato plants and in the corresponding tuber sprouts in both the 2021 and 2022 samples (R = 0.74, *p* = 0.036, for both 2021 and 2022), which may indicate that the efficiency of PVY transmission through tubers is directly linked to the level of the virus infection in leaves at the time when tubers are formed.

No connection was observed between PVY levels and geographic location for both leaves and tubers. While the virus levels in the field-grown plants were higher in the Moscow region compared to the Astrakhan region in the year 2021, it was reversed in the following year.

### 2.2. Phylogeny of De Novo-Assembled PVY Genomes

The alignment of the NGS libraries to the reference containing all known types of PVY [11] showed that all PVY reads in our samples originated from the parental PVY-Eu-N and PVY-O, which together covered the entire length of the PVY genome (Figure 3). In most libraries, the coverage was several thousands of nucleotides in depth; the lowest coverage was at least several hundred across the entire length of the PVY genome (Figure 3), making it possible to carry out the de novo assembly of full-length viral contigs.

The NGS libraries were used for de novo viral genome assembly. For further analysis, we considered only nearly full-length contigs (n = 94) which should include the entire protein coding region. The nucleotide identity between the assembled contigs ranged from 92.5% to 100.0% (average ± SD, 97.4 ± 2.1%), with only two new contigs being completely identical. The amino acid identity of those contigs ranged from 95.9% to 100.0% (average ± SD, 98.5 ± 1.0%; Appendix A). We carried out a phylogenetic analysis of the de novo PVY contigs which were produced together with a representative set of the full-length genomic sequences of all identified PVY types and their recombinants [11]. We found that our contigs belonged to the previously discovered types of PVY, including PVY-NTNa, PVY-NTNb, PVY-N:O, PVY-SYR-I, PVY-SYR-II, PVY-SYR-III, and PVY-261−4, with PVY-NTNa and PVY-NTNb being the most widespread (Figure 4). Genomes of all these PVY types were recombinants between PVY-O and PVY-N, and the arrangements of the PVY-O- and PVY-N-derived sections in their genomes were in good agreement with the pattern of distribution of the PVY-N- and PVY-O-type reads in the NGS libraries (Figure 3). In particular, in all our libraries, there was exclusively PVY-N-type coverage in the region 1000–2390 nt (HC-Pro gene), PVY-O-type coverage in the region 2390 to 5800 nt (P3 and CI genes), and PVY-O type coverage in the short 3′ proximal genome section approximately from the position 9500 (Figure 3 and Figure 5a). The rest of the PVY genome had a mixed PVY-O- and PVY-N-type coverage with varying proportions in different libraries. While in the most libraries the section 5800 to 9500 had higher PVY-N coverage, supporting our finding that PVY-NTNa and PVY-NTNb contigs were the most common, the continuous PVY-O-type coverage in this region was consistent with the presence of PVY-N:O and PVY-Wi contigs, which have PVY-O-type genomes from the position 3390 nt to the 3′ end (Figure 3 and Figure 5a).

### 2.3. Factors Affecting the Diversity of PVY

PVY has a number of isolates, many of which are recombinants between “parental” PVY-N and PVY-O types (Figure 5a, Appendix A). Recombinant PVY variants may have multiple recombination points and highly conserved genome regions (for example, genome section approximately from 700 to 5800, Figure 5a), making it essential to have knowledge of complete genome sequence to assign PVY genome to a certain isolate. Therefore, in this study, we used full-length de novo-assembled PVY genomes to compare PVY complexes associated with different conditions.

By analyzing full-length de novo PVY contigs (n = 94) originated from potato plants in two regions of Russia, we identified PVY isolates belonging to eight types (Figure 5a, Appendix A). We further explored the possible effects of geographic location, developmental stage (leaves from mature plants and sprouts from the tubers produced from infected plants), and the variety of host potato plants.

We analyzed the regional richness of PVY isolates and found Shannon’s diversity index was significantly higher in the Astrakhan region (1.759) than in the Moscow region (0.8617), *p* = 0.00000769 (Figure 5c). In both regions, the most abundant isolates were PVY-NTNa and PVY-NTNb, which accounted for 92.5% contigs (37 of 40) in the Moscow region and for 57.4% contigs (31 of 54) in the Astrakhan region. The proportion of PVY-NTNa or PVY-NTNb contigs among all contigs in the Moscow region was significantly higher, *p* = 0.0005953 (Fisher’s exact probability test, two tailed *p*-value, Figure 5b).

Although PVY-NTNa and PVY-NTNb were dominant isolates in both regions (Figure 5b), there was a regional difference in their proportion. We found that PVY-NTNa was significantly more abundant in the Moscow region (28 NTNa and 9 NTNb contigs) compared to the Astrakhan region (10 NTNa and 21 NTNb contigs), *p* = 0.000523 (Fisher’s exact probability test, two-tailed *p*-value).

We did not observe a statistically significant difference between the diversity of PVY isolates in the sprouts from the tubers produced by infected plants (n = 8) compared to those in the leaves of parental field-grown plants (n = 6) (Figure 5b). In some cases, the composition of PVY isolates in “leaves” and their respective “tubers” differed considerably. For example, in the leaf samples of the Indigo variety grown in the Astrakhan region in the year 2021, we identified three types of PVY, but the sprouts for their progeny tubers had eight PVY variants, which might indicate that vertical transmission through tubers has an effect on the diversity of PVY isolates.

We tested whether the degree of susceptibility of potato variety to PVY has an effect on the number of PVY isolates infecting this potato variety. We found that, although Indigo plants had higher levels of PVY infection compared to La Strada (Figure 2), there was no significant difference between the number of PVY isolate types found in Indigo plants (all eight PVY types found in our study) compared to La Strada (six of eight PVY types found in our study, Figure 5b).

## 3. Discussion

We investigated the genetic diversity of PVY vectored by aphids in different climate zones of Russia and the effect of the subsequent transmission of the virus to the next vegetative generation of potatoes via tubers. The genetics of PVY could be potentially affected by the aphid vectors, by reservoir plants in which PVY could survive during winters when potato plants are not available, and by the selection of certain variants during vertical transmission through tubers. Our experiments included planting virus-free seed potatoes of two commercial potato varieties with different levels of susceptibility to PVY, Indigo and La Strada, in field conditions in the Moscow and Astrakhan regions of Russia. The Astrakhan region, compared to the Moscow region, has much warmer summers, as well as shorter and milder winters, which, according to previous reports, result in an increased density of aphids vectoring PVY [23]. The potato plants were grown in the fields without the application of aphid control measures to allow for the transmission of aphid-vectored viruses, including PVY, circulating in these regions. The progeny tubers produced by the aphid-exposed plants were sprouted in insect-proof conditions. Leaves from the field-grown plants and the tuber sprouts were analyzed using high-throughput RNA-seq to comprehensively characterize PVY populations (Figure 1). The alignment of the NGS reads to the sequences of all known isolates of PVY showed the presence of PVY only in the samples from the fully grown field plants, but not in the newly grown leaves. We found that all PVY NGS reads identified in our libraries belonged to the PVY-N and PVY–O types, which are widely regarded as “parental” PVY types [11]. There was no full genome coverage of parental N or O types in any of the libraries (Figure 3), indicating the presence of PVY genomes of “recombinant” types. To identify individual PVY genomes in the samples, we assembled a series of de novo PVY contigs using the high-throughput sequencing libraries with several de novo assemblers to achieve the best results. The accuracy of our approach was confirmed by the generation of de novo PVY contigs belonging to previously identified PVY types (Figure 4). Also, a good agreement between the distribution of PVY-N- and PVY-O-derived sections in the assembled contigs (Figure 5), and the overall distribution of the reads derived from these parental strains in the NGS libraries (Figure 3), further support the correctness of our analysis. We identified the major variants of PVY characteristics for the Moscow and Astrakhan regions in two consecutive years in both leaf and tuber samples (Figure 5, Appendix A).

In the Moscow region, we detected recombinant variants PVY-NTNa, PVY-NTNb, PVY-N:O, and PVY-N-Wi in leaf samples, while in the samples from the progeny tubers PVY-NTNa, PVY-NTNb, and PVY-SYR-I were found (Figure 5b). In the Astrakhan region, the leaf samples contained PVY-NTNa, PVY-NTNb, and PVY-SYR-III, while the corresponding tuber samples had a wide variety of variants, including PVY-NTNa, PVY-NTNb, PVY-N:O, PVY-N-Wi, PVY-SYR-I, PVY-SYR-II, PVY-SYR-III, and PVY-261-4. In general, the distribution of the PVY-O- and PVY-N-derived sections in the contigs was in good agreement with the distribution of the NGS reads of the parental types (Figure 3). However, it cannot be excluded that additional minor recombinant PVY variants with novel arrangements of the O- and N-type sections in the region from 5800 to 9100 could be present. In all libraries, we only observed PVY-N-type reads in the regions 700 to 2390 (HC-Pro gene), and exclusively PVY-O-type reads in the section 2390-5850 (P3—CI genes), which agrees well with the genetic structure of all PVY types detected (Figure 5a). The 5′ terminal genomic regions (1 to 700) and the 3′ half of the genome (from position 5850 to the 3′ terminus) had a mixed coverage of both PVY-O- and PVY-N-type reads, indicating the presence of multiple PVY variants generated as a result of recombination events between parental PVY-O and PVY-N types. Notably, our study did not detect complete parental PVY-O and PVY-N (Eu-N). Also, we found a very high, approximately 99.8%, nucleotide identity of some PVY isolates from other European countries. This strongly suggests that the recombinant genomes discovered in Russia were most likely generated previously in different geographic locations. Importantly, both PVY-N and PVY-O types are common in Europe, as shown by a recent Irish report which identified PVY-Eu-N and PVY-O alongside the recombinants PVY-NTNa and PVY-N-Wi [24]. In this respect, the range of PVY isolates in Russia, in particular in the Astrakhan region, was more similar to that in Israel, where PVY-NTNa was predominant with considerable proportions of PVY-N-Wi and PVY-SYR-III. Similar to Russia, no PVY-Eu-N was found and PVY-O was extremely rare [25].

A displacement of parental PVY-N and PVY-O types by their recombinants was observed in different geographic locations, suggesting that the recombinant PVY variants, such as the NTNa and NTNb types which accounted for two thirds (68 of 100) of the contigs assembled in this study, could be more competitive than their parental isolates. The mechanisms of such displacement in the case of PVY are not known, but the appearance of recombinant RNA virus isolates which almost replace parental types of viruses was also reported for animal-positive strand RNA viruses [26,27]. It is possible that recombinant PVY genomes belonging to the same recombinant type might be generated from parental PVY-O and PVY-N types on several occasions as a result of independent recombination events. This could lead to the variation of the recombination points [27] and/or combine genome sections of the same type that are evolutionary distinct, as was observed for the generation of a recombinant honey bee picorna-like virus in different locations [28].

We tested whether the diversity of PVY isolates was influenced by the geography, the type of plants (leaves from mature plants or sprouts from tubers), and the susceptibility of the host potato variety to PVY. We found that the Astrakhan region had a significantly higher richness of PVY isolates (Figure 5c). This could be explained by a higher number of aphid vector species of the virus due to warmer summer weather, as well as shorter and milder winters, that could increase the chances of survival of viruliferous aphids, and a higher diversity of plants which may act as reservoirs for PVY. Also, a milder climate in the Astrakhan region could support a wider diversity of reservoir plants where PVY could survive during the winter. Indeed, in the Moscow region, the fields with experimental potato plants were surrounded by urban areas and cultivated fields where insecticides were routinely applied, while in the Astrakhan region, the experimental fields were in the immediate vicinity of wild-plant-species-rich natural steppe and riparian areas which were free of insecticide applications, as well as small holdings growing potato, tomato, and pepper. It is possible that the availability of a wider range of reservoir plant species may favor the selection of a wider variety of PVY variants, because different PVY variants may be better adapted to certain reservoir plants.

Our study showed that the vertical transmission of PVY to the next vegetative generation populations via the tubers of PVY did not have a significant impact on the composition of PVY complexes. For the entire experiment, we observed similar numbers of PVY types in the leaves of field-grown plants (n = 8) and in the sprouts of the tubers collected from the PVY-infected field potato plants (n = 11), but the PVY strain compositions in the leaves and the sprouts from corresponding tubers showed that differences were often not the same (Figure 5b). This might indicate that tuber transmission could potentially lead to changes in PVY composition structure, which can be explained by the acquisition of the additional isolates of PVY between the leaf sample and tuber collections, or by the effect of tuber transmission (for example, through the genetic bottleneck effect). Indeed, the highest number of variants (n = 10) was found in the Indigo tuber sprouts sample from Astrakhan.

We tested whether the degree of susceptibility of potato variety to PVY has an effect on the number of PVY isolates infecting this potato variety. We found that, although Indigo plants had higher levels of PVY infection compared to La Strada (Figure 2), there was no significant difference between the number of PVY isolate types found in Indigo plants (all 12 PVY types found in our study) compared to La Strada (6 of 8 PVY types found in our study).

Our study for the first time provided data on the genetic composition of PVY circulating in the Moscow and Astrakhan regions of Russia via the high-throughput sequencing of RNA populations from potato plants naturally infected by the aphid vectors and from the tubers produced by the field-infected plants. A number of PVY types were identified, including PVY-NTNa, PVY-NTNb, PVY-N:O, PVY-N-Wi, PVY-SYR-I, PVY-SYR-II, PVY-SYR-III, and PVY-261-4. All PVY variants identified in Russia were recombinants composed of the genome sections of PVY-N and PVY-O types. There was a complete absence of the isolates with full-length “parental” sequence types, such as PVY-N type (PVY-Eu-N) and PVY-O. We found a significantly higher diversity of PVY types in the Astrakhan region compared to the Moscow region, possibly because shorter winters and warmer summers in the Astrakhan region support greater numbers of aphid vectors and a wider variety of potential PVY reservoir plants. The composition of the PVY variants in the tuber sprouts was not always the same as in their parental plants, suggesting that tuber transmission has an impact on the genetics of PVY.

## 4. Materials and Methods

### 4.1. Plants and Experimental Design

Virus-free seed potatoes of the varieties Indigo were developed by Doka-Gene Technologies LLC (Rogachevo, Russia), and La Strada were developed by Cygnet Potato Breeders Ltd. (Scotland, UK). Both potato varieties are PVY-susceptible. Approximately up to 50% of Indigo plants and up to 25% of La Strada plants are typically infected with the virus [29]. The seed potatoes were derived from the plants produced from apical meristem tissue culture in aphid-proof greenhouse conditions excluding infection with potato viruses (Doka-Gene Technologies LLC, Rogachevo, Moscow region, Russia). The absence of viruses, including PVY in seed potato sprouts, was confirmed with qRT-PCR tests.

The seed potatoes were planted in the fields in the Astrakhan region on 6 April 2021 and 1 April 2022, and in the Moscow region on 20 May 2021 and 11 May 2022, with 50 to 60 seed tubers of each variety being planted each year in each location. Leaf sample collections from the mature plants were carried out in the Astrakhan region on 20 June 2021 and on 3 July 2022 and in the Moscow region on 28 July 2021 and 16 August 2022. The first collection of newly sprouted leaves in the Moscow region was carried out on 14–17 June 2022.

Field-produced tubers were collected in the Astrakhan region in the first week of August 2021 and 2022, and in the Moscow region on 9 September 2021 and 6 September 2022. The tubers were stored for 6 months (for the year 2021) and for 2 months (for the year 2022) at +4 °C, then the apical buds were excised from the tubers; the plantlets were grown in an insect-free greenhouse at 21–22 °C and were sampled at the 5 to 8 leaf stages 3–4 weeks after planting.

The Astrakhan and Moscow regions had significantly different climates. In the years when the experiments were conducted, the Astrakhan regions had two winter months with average temperatures below freezing (−2.1 °C and −2.2 °C) for these months in the winters 2021/2022 and 2020/2021, correspondingly; while the Moscow region had 4-month and 5-month periods with average temperatures below freezing (−7.4 °C and −4.6 °C) for these months in the winters 2020/2021 and 2021/2022, correspondingly. Average temperatures from May to August for the years 2021 and 2022 were +18.1 °C and +16.9 °C in the Moscow region, and 25.9 °C and 23.4 °C in the Astrakhan region.

### 4.2. RNA Extraction and High-Throughput Sequencing

Leaf and tuber sprout samples were frozen with liquid nitrogen, homogenized with pestle and mortar, and combined to prepare pools of 10 to 25 field plants and 10–12 tubers of the same treatment. Total RNA was extracted with the TRIzol reagent (Invitrogen, ThermoFisher Scientific, Waltham, MA, USA) as recommended by the manufacturer. The samples were treated with DNase I (DNase I, RNase-free, HC (50 U/µL)—Thermo #EN0523) and re-precipitated with trizol-chloroform. Then, the quantity and quality of the RNA preparations were assessed using NanoDrop ND-1000 (Nanodrop Technologies, Wilmington, DE, USA) and agarose gel electrophoresis, respectively. The high-quality RNA was subsequently used for RNA-seq.

The polyadenylated fraction of total RNA extracts were sequenced using the Illumina NovaSeq 6000 platform via CeGaT GmbH to obtain RNAseq libraries (n = 42) containing 26,862,043 to 124,969,530 reads (average ± SD, 4,141,652 ± 22,843,195) of 2 × 100 nt pair-end reads with Q30 ranging from 84.43 to 95.75%. The demultiplexing of reads was performed using Illumina bcl2fastq v2.20 via CeGaT GmbH. Adapters were removed with the use of Skewer v0.2.2 [30] via CeGaT GmbH. Read quality was assessed through the means of FastQC v0.11.9 [31] and SeqKit v2.3.0 tools [32]. The RNA-seq libraries with removed adapters were deposited in the Sequence Read Archive (SRA), BioProject accession number PRJNA1002225 (Appendix A).

### 4.3. De Novo Assembly and Sequence Sources

The de novo assembly was performed using three algorithms: Trinity v2.14.0 (standard k-mer length, 25 nt), rnaSPAdes v3.15.4 (standard k-mers: 33, 49 nt), and rnaviralSPAdes v3.15.4 (standard k-mers: 21, 33, 49 nt) [33,34,35]. Assembled contigs statistics were calculated with QUAST v5.2.0 software [36].

A total of 9,873,063 contigs were obtained, of which 897 with the length of >9000 nucleotides were attributed to PVY based on the similarity with the representative set of PVY genomes revealed via BLAST search [37]. We identified 100 contigs containing full-length genomic RNA sequences of PVY, which were chosen for further analysis and were separated from others using Python v3.11.3 author script. The translation of the assembled nucleotide sequences was carried out with NCBI ORFfinder v0.4.3. All full-length contigs contained an extended open reading frame covering most of the genome. The taxonomic classification of nucleotide and translated protein sequences was carried out using the BLAST algorithm (BLASTn v2.14.0+) and the NCBI database. To evaluate the assembly quality and correct possible errors, the original reads were mapped to the resulting assembly. Mapping was carried out using HISAT2 v2.1.0 [38]. Samtools package v1.10 was used for operations with sam/bam files [39]. The assembly was checked for single nucleotide errors, short insertions, deletions, and breakpoints (stacks of soft- and hard-clipped reads) using Tablet program v1.19.09.03 [40].

The sequences of selected full-length de novo PVY contigs were deposited to GenBank under accession numbers OR479972—OR480071. A summary of the samples, RNA-seq libraries and contigs is shown in Appendix A.

Among the assembled contigs, we also identified those of other potato viruses (potato virus M and potato virus S), as well as oomycete (*Phytophthora* sp.), fungus (*Fusarium* sp.), and bacterium (*Ralstonia solanacearum*, *Clavibacter michiganensis*) pathogens of potato. The numbers of these contigs identified in the libraries are shown in Appendix A.

### 4.4. Phylogenetic and Comparative Analysis of PVY Sequences

The multiple alignments of the 246 nucleotide and amino acid sequences (152 PVY isolates from GenBank and 94 complete de novo PVY contigs obtained in this study) were performed independently in the ClustalO v1.2.3 [41] and Muscle v3.8.1551 [42] programs using default settings. To eliminate poorly aligned and diverged regions, Gblocks v0.91b was used with the default parameters and the resulting fragments were concatenated [43]. The 5′ and 3′ ends of the sequences were trimmed to create a ‘complete ORF’ alignment. All full-length contigs had sufficient uninterrupted read coverage of their respective NGS libraries.

The distance matrix of amino acid sequences was calculated using an R package bio3d v2.4.4 [44]. Distance matrices of both the nucleotide and the amino acid sequences can be found in the Supplementary section of this article (Appendix A). The heatmap representation of the distance matrix was constructed using Python v3.11.3 author script (Appendix A). The same procedure was applied to the translated protein sequences (Appendix A).

The maximum-likelihood (ML) trees built using the best-fit models were found to be GTR+G+I for nucleotide sequences and amino acid sequences within MEGA-X software v10.2.4 [45] with 1000 bootstrap replicates to determine statistical significance. The trees were visualized using iTOL v6.8 [46].

### 4.5. Statistical Analysis

Statistical analyses were conducted using R Software version 4.2.1 (R Foundation for Statistical Computing, Vienna, Austria) [47]. Analysis of variance (ANOVA) was used to determine the significance of differences between virus read numbers in NGS libraries. Fisher’s exact probability test for 2 × 2 contingency tables was used to assess significance of differences in proportions of virus variants.

## Figures and Tables

**Figure 1 ijms-24-14833-f001:**
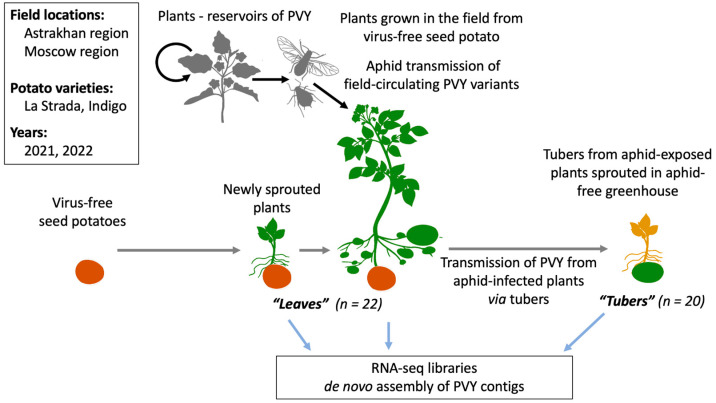
Experimental design and sampling. Virus-free seed potatoes were planted in the fields without aphid control to allow for the aphid-mediated transmission of PVY from reservoir plants in the Astrakhan and Moscow regions. The tubers produced in the field conditions were sprouted in an aphid-proof greenhouse. The leaves collected from the field-grown plants (“Leaves”), and the sprouts from tubers (“Tubers”) were used for RNA-seq.

**Figure 2 ijms-24-14833-f002:**
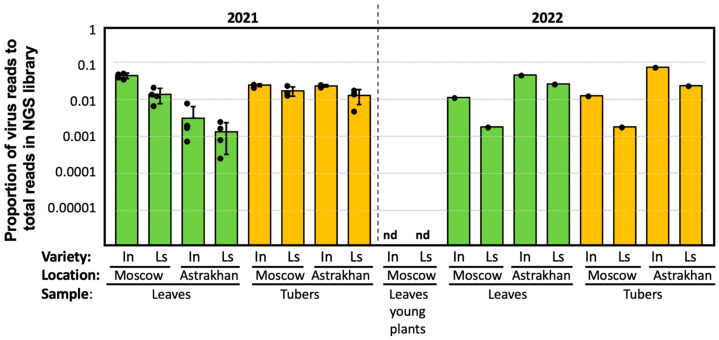
Development of PVY infection in field-grown potato plants (green bars) and in the sprouts from the field-produced tubers (yellow bars). Bars indicate average proportions of the PVY reads in the NGS libraries of each treatment, error bars show standard deviation, black dots indicate proportions of PVY in individual libraries, the year is indicated above the graph. The NGS libraries are described in detail in Appendix A. Sample labels indicate type (“Leaves”—field-grown plants, 10–12 weeks after tuber planting, “Leaves young plants”—field-grown plants, 5 weeks after tuber planting, “Tubers”—sprouts from field-produced tubers); potato varieties: In—Indigo, Ls—La Strada.

**Figure 3 ijms-24-14833-f003:**
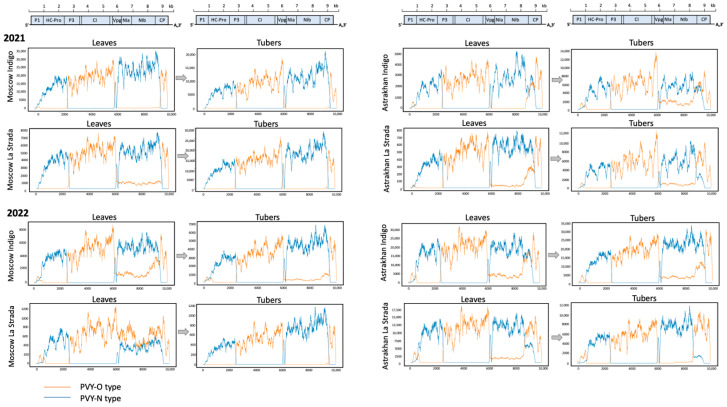
Coverage plots and distribution of the PVY-O-type (orange graph) and PVY-N-type (blue graph) reads in the PVY populations. Genetic maps of PVY are shown above the coverage panels.

**Figure 4 ijms-24-14833-f004:**
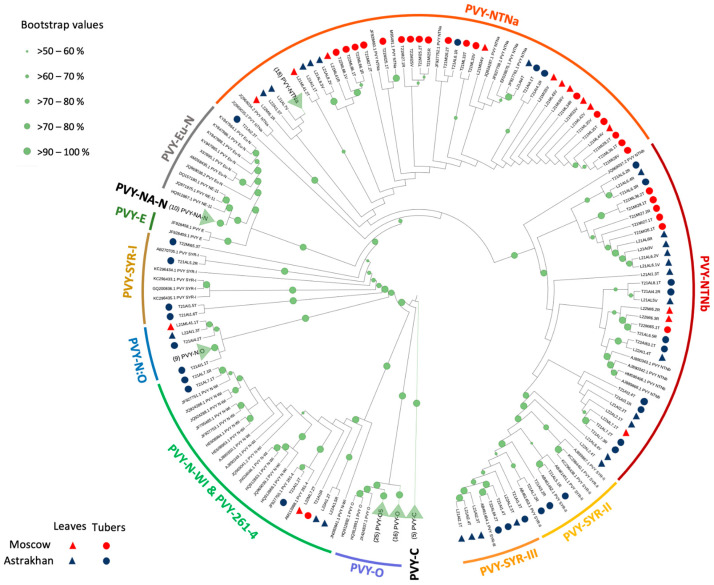
Phylogenetic relationship between de novo PVY contigs identified in Russia and previously characterized PVY isolates. Maximum-likelihood phylogenetic trees were generated for complete nucleotide sequences. Bootstrap values for 1000 replicates are indicated by the size of green circles on branches (the smallest 50–60%, the largest 90–100%). Tip labels for the previously characterized PVY isolates show the GenBank accession number and the PVY type. De novo contigs are marked with triangles or circles as described in the figure. The sizes of green circles at the nodes indicate bootstrap values as described in the figure. Tip labels for de novo contigs show type of sample (first letter, L—leaves, T—tubers), year of collection (2021–2021, 2022–2022), and region and potato variety (AI, AL—Astrakhan Indigo or La Strada; MI, ML—Moscow Indigo or La Strada).

**Figure 5 ijms-24-14833-f005:**
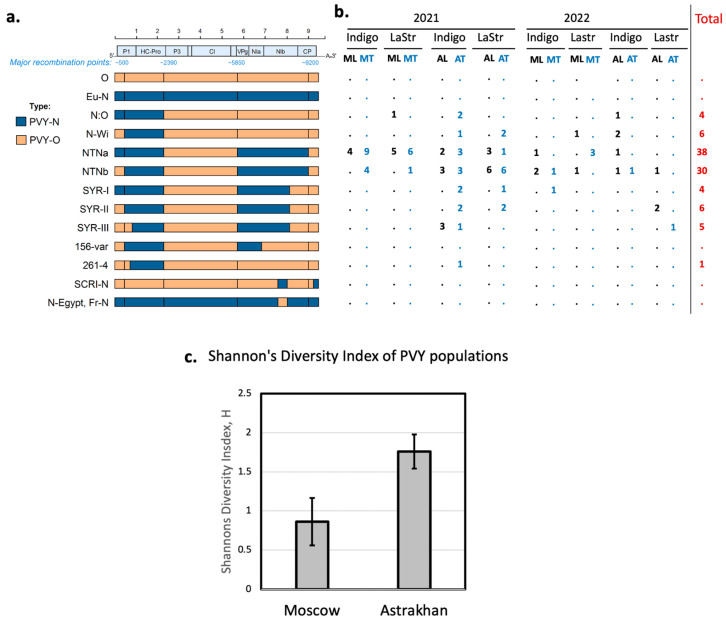
Recombinant variants of PVY identified in the Moscow and Astrakhan regions of Russia. (**a**) Shown are genome structures of the previously characterized major recombinant variants of PVY. The sections of genomes of the PVY-N- and PVY-O-derived genome sections are shown as dark blue and orange rectangles, correspondingly. Approximate nucleotide positions of the major previously known recombination points are shown in blue above. (**b**) Number of contigs assembled in this study. Shown are year of sampling; potato varieties, Indigo, LaStr—La Strada; and origin and type of samples, AL—Astrakhan region, leaves, AT—Astrakhan region, tubers, ML—Moscow region, leaves, MT—Moscow region, tubers. (**c**) Regional richness of PVY isolates. Significantly higher Shannon’s diversity index of PVY complexes in the Astrakhan region compared to the Moscow region (*p* = 0.00000769), error bar indicates 95% confidence intervals (CI).

## Data Availability

The data supporting the findings of this study are available within the article and in the NCBI database, BioProject accession PRJNA1002225.

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
