# Peer review of "The Temporal and Geographical Dynamics of Potato Virus Y Diversity in Russia"

_ijms, 2023, doi:10.3390/ijms241914833_

Round 1
Reviewer 1 Report
The manuscript “Temporal and geographical dynamics of Potato virus Y diversity in Russia” by Viktoriya and colleagues here describe using high-throughput sequencing to investigate PVY populations transmitted to potato plants by aphids in two different regions of Russia and investigate the impact of tuber transmission on PVY genetics. They finally found that a higher diversity of PVY in the warmer region and tuber transmission could impact on PVY genetics.
This manuscript is well written but some experiments is not well designed. There are some comments listed below need address and discussion:
1. How to rule out the possibility of those recombinants were not resulted from the incorrect viral genome assembly? For example, If there are two PVY types co-infection in the same leaf sample, this two genome types might be assembled into the same genome, and might be recognized as recombinant. To rule out this kind of errors, need perform RT-PCR to confirm those recombinants, at least in those new types of recombinant variants.
2. in the experiment, the original types of PVY variants from Astrakhan region and Moscow region reservoir plants is unknown and uncontrolled, while they were transmitted to potato by aphid and found there are more recombinant variants in the Astrakhan region than the Moscow region, these results might be generated for original variety of field virus types as inoculation resources.
3. Line 28: please check four types or five types? It listed five types NTNa, NTNb, N:O, N-Wi, and SYR-1
4. Line 152: add blank space at the start of paragraph
5. Line 160: lengths change to length
6. Line 224: why all novel recombinants had PVY-N type HC-Pro gene and PVY-O type P3-CI gene block, it needs discussion in the discussion parts
Author Response
1. Answer: We thank Reviewer 1 for the constructive comments. We fully appreciated that de novo assembly of virus sequences may result in artifacts therefore we used three de-novo assembling algorithms with different k-mer lengths (Trinity v2.14.0 (standard k-mer length, 25 nt), rnaSPAdes v3.15.4 (standard k-mers: 33, 49 nt), rnaviralSPAdes v3.15.4 (standard k-mers: 21, 33, 49 nt) (See section 4.3 “De novo assembly and sequence sources”, L. 468-489). We also detected individual 100 nr NGS reads covering the PVY-O / PVY-N junctions from the NGS libraries where the recombinants were identified, the 5’ and 3’ halves of which showed identity to either PVY-O type or PVY-N type.
Also, the assembly of many recombinant variants (such as PVY-NTNa, PVY-NTNb, etc), which were nearly identical to those found in previous studies, further confirms accuracy of our de novo assembly pipeline.
We agree with the Reviewer’s suggestion that additional sequencing data (long RT-PCR fragments) should be presented to demonstrate the presence of novel types of recombinants between PVY-O and PVY-N types. Importantly it would be necessary to analyze additional potato samples to determine whether the novel recombinants persist in potato plants and remain in the tuber progeny in the next season. Such additional analysis would take several months, therefore we decided to present in this MS only the contigs of the major PVY variants (n=94) which we assembled in this study. We therefore deleted MS parts describing potential novel recombinants (6 contigs of 100 presented in the original submission). We noted (lines 294-297) that “ … It cannot be, however, excluded that additional minor recombinant PVY variants with novel arrangements of the O- and N-type sections in the region from 5800 to 9100 could be present.”.
2. Answer: Yes, several types of PVY recombinants might be generated in different reservoir plant species. The aim of this study was to investigate distribution of PVY variants which are naturally transmitted by aphids from reservoir plants to potato plants in two climate regions of Russia. Finding out natural reservoir plants for PVY in Moscow and Astrakhan regions is important and could be carried out in the future with the information on PVY diversity determined in this study.
3. Answer: Thank you, corrected as suggested.
4. Answer: Thank you, corrected as suggested.
5. Answer: Thank you, corrected as suggested.
6. Answer: The block of PVY-N type HC-Pro and PVY-O type P3 CI genes is found in all PVY variant founding Astrakhan and Moscow (including previously characterized most abundant PVY-NTNa, -NTNb, -NO, N-Wi, -SYR…). Moreover, these recombinants are common worldwide (see ref Green et al. our Reference 11) It is not clear why PVY genomes with this recombinant overcompete parental variants and answering this question would require additional experiments beyond the aims of this study.
According to computational analysis of PVY genomes (Hu et al., 2009b) it was suggested that the number of recombinant patterns reported for PVY isolates from potato are relatively limited, and the positions of the main recombinant junctions (RJs) are remarkably conserved, although it is as yet unclear why. There are no special sequences or RNA secondary structures found associated with the most common RJs. Interesting that hypervariable areas map to most of the RJs. Thus, potyviruses contain fixed hypervariable areas and RJs in key parts of the genome which provide mutational robustness which may be potentially involved in host adaptation.
Reviewer 2 Report
I had the honor to read the manuscript titled Temporal and geographical dynamics of Potato virus Y diversity in Russia.
The manuscript is well written. Data are clearly described and are consistent. The topic is interesting and to my opinion could have important feedbacks.
What I strongly suggest to the authors is to revise and edit the experimental design because is not fully linear and could bring to some misunderstandings concerning timing and samplings.
I would recommend this manuscript to be published, after a minor revision.
Regards
Author Response
Answer: We are very pleased that Reviewer 2 found our MS interesting and recommended to publish. We understand that in general, the Reviewer agrees that experimental design was appropriate, and only concerns are related to the timing and sampling. We should note that although our sample experiment was carried out over two years, all address analysis was done simultaneously for both years. Therefore this is reflected in the manuscript sections.
Reviewer 3 Report
The authors adopted the RNA-Seq technology to investigate the PVY diversity in different climate zones in Russia. New PVY recombinants were identified. The experimental design is good and was conducted over a long-time period which provides better insights. However, I have some major/minor comments to the authors to consider.
Major comments:
Given that your study is conducted under field conditions, the next comments are important to consider:
Although the focus of the study is PVY, however the RNA-Seq data would provide you with a wealth of data about the other viruses that may co-exist. The effects of other potato viruses should not be ignored and may provide better explanation of the obtained results. At least the major identified potato viruses should be listed.
Are there any previous reports of synergistic/antagonistic relationships between the identified potato viruses and PVY?
Furthermore, there is no description of the other pathogens (ex: bacteria, fungi,….etc.) that co-exist with PVY during infection. At least the major pathogens detected in the RNA-Seq data should be listed. Are there any pathogens associated with biological control and may shape the diversity of detected microbes?
Minor Comments:
English language needs revision.
4.6. statistical analysis>> list the name(s) of statistical tests conducted in your study.
Line 359>> Correct the reference style.
English language needs revision.
Author Response
Major comments:
Answer: We found Potato virus M and Potato virus S contigs assembled in 21 and 2 NGS libraries respectively (Supplementary Table S1), and described this in “Materials and Methods” (lines 450-453). “Among the assembled contigs we also identified those of other potato viruses (potato virus M and potato virus S), as well as oomycete (Phytophthora sp), fungus (Fusarium sp.) and bacteria (Ralstonia solanacearum, Clavibacter michiganensis) pathogens of potato. The numbers of these contigs identified in the libraries are shown in Supplementary Table S1. “
It should be noted that our study was not designed to investigate interactions between different viruses, or oomycetal, fungal, or bacterial pathogens within individual plants. The aim was to investigate the diversity of PVY, therefore we used pools from 10 to 25 plants, rather than individual plant sampling, which would be more appropriate to investigate interaction between different viruses.
Although there are reports on synergistic interactions between PVY and Potato virus S (PVS) and Potato virus M (PVM) (for example Hameed A, Iqbal Z, Asad S, Mansoor S. (2014) Detection of Multiple Potato Viruses in the Field Suggests Synergistic Interactions among Potato Viruses in Pakistan. Plant Pathol J. 30(4):407-15. doi: 10.5423/PPJ.OA.05.2014.0039 ), the viruses which were detected in several our libraries (Materials and methods, lines 446-449), we did not investigated possible synergism between PVY and PVM, or PVY and PVM+PVS because we used pooled samples composed of 10 to 25 plants. We could not be sure that PVY, PVM and PVS were present in the same plants.
Minor Comments:
English language needs revision.
Answer: We re-checked English language with a native English speaking biologist.
4.6. statistical analysis>> list the name(s) of statistical tests conducted in your study.
Answer: We provided an additional explanation in ‘Materials and Methods’, (lines 470-473) “...Analysis of variance (ANOVA) was used to determine significance of differences between virus read numbers in NGS libraries. Fisher’s exact probability test for 2x2 contingency tables was used to assess significance to differences in proportions of virus variants. “
Line 359>> Correct the reference style.
Answer: Corrected, Moore et al 2011 => Ref 27.
Reviewer 4 Report
Authors Viktoriya Samarskaya and coworkers presented here a manuscript entitled
„Temporal and geographical dynamics of Potato virus Y diversity in Russia.“
The authors used high-throughput sequencing to analyse potato virus Y (PVY) populations in two regions of Russia. All virus isolates identified were recombinations, including novel types.
The manuscript is comprehensive and well written. I can recommend it for publication in IJMS, but after addressing some minor issues listed below.
Suggestions for further improvement of the manuscript:
Line 83: naive – should be native
Lines 84-85: different ranges of aphid vectors – please specify
Line 129: ... levels of PVY were 1.4 to 6.2-fold higher in the leaf material from fully grown Indigo... – comparison the percentage of virus reads in libraries does not give a true picture of virus concentration in plant tissues. The standard method for this task is qRT-PCR.
Line 302: the tuber sprouts which were analyzed – delete the word „which“
Line 424: approximately 50% of Indigo plants, and 25% of La Strada plants could be infected – why 50% or 25% of the plants? Are the varieties genetically heterogeneous?
Line 427: use full stop at end of sentence
Line 447: yearс – should be years
Line 473: ... contigs were obtained, nucleotides, of which ...– does not make sense
Some minor issues found, listed above.
Author Response
Answer: We are very pleased that Reviewer 4 found our MS comprehensive and well written and recommended to publish it after addressing minor points.
Suggestions for further improvement of the manuscript:
Line 83: naive – should be native
Answer: We used potato plants which were previously infected with PVY, i.e. which were naive (naïve) meaning that these plants did not encounter PVY before. We agreed that this characteristic of plants could be confusing and replaced it with “virus-free” in the abstract (line 21) and throughout the text (line 84).
Lines 84-85: different ranges of aphid vectors – please specify
Answer: We deleted this sentence because we don’t have enough data of aphid species range and density for the year 2021 and 2022 years. We cited the paper Fominykh et al (2017), Monitoring of Potato Viral Diseases in the Pskov and Astrakhan Regions of Russia. Plant Prot. Vestn. 2017, 4 (94), 29–34 (our Reference 23, line 265), which showed that Aphis fabae Scop., Aphis nasturtii Kalt., Aulacorthum solani Kalt.,Macrosiphum euphorbiae Thomas, Myzus persicae Sulz, which could be vectors for PVY were detected in the region with the climate similar to that in the Moscow region (Pskov region) and the Astrakhan region.
Line 129: ... levels of PVY were 1.4 to 6.2-fold higher in the leaf material from fully grown Indigo... – comparison the percentage of virus reads in libraries does not give a true picture of virus concentration in plant tissues. The standard method for this task is qRT-PCR.
Answer: We agree that qPCR could be more accurate therefore we specified that we observed a higher number of NGS reads (Line 131)“ Indeed, we found that the number of of PVY reads were 1.4 to 6.2-fold higher in the in the NGS libraries from the leaf material from fully grown Indigo plants compared to that from La Strada in both years, and this difference was statistically significant for the 2021 Moscow region samples (p = 0.0007, ANOVA).“
It was important to compare relative levels PVY in the NGS libraries. We universally applied the same method of quantification (percentage of NGS counts) across all samples, therefore we believe that it is possible to state that the levels of PVY in Indigo were higher.
Estimation of quantification of virus levels by comparing proportion of virus reads is widely used (for example https://journals.plos.org/plospathogens/article?id=10.1371/journal.ppat.1004230 Fig. 7A).
Line 302: the tuber sprouts which were analyzed – delete the word „which“
Answer: Thank you, corrected as suggested.
Line 424: approximately 50% of Indigo plants, and 25% of La Strada plants could be infected – why 50% or 25% of the plants? Are the varieties genetically heterogeneous?
We specified (Lines 381-383) that “up to 50%” and “up to 25%” of Indigo and LaStrada plants could be infected with PVY according to the variety description (Reference 29) . We don't have data on genetic diversity of plants used in our experiment.
Line 427: use full stop at end of sentence
Answer: Thank you, corrected as suggested.
Line 447: yearс – should be years
Answer: Thank you, corrected as suggested.
Line 473: ... contigs were obtained, nucleotides, of which ...– does not make sense
Answer: Thank you, corrected to “ A total of 9,873,063 contigs were obtained…” (line 432).
Round 2
Reviewer 1 Report
no more comments
Reviewer 3 Report
The authors addressed all my concerns and I recommend the manuscript for publication.